# Eating Patterns among Emergency Medical Service Providers in the United States: A Qualitative Interview Study

**DOI:** 10.3390/nu14224884

**Published:** 2022-11-18

**Authors:** Tegan Mansouri, George Ghanatios, Lori Hatzinger, Rachel Barich, Ebriama Dampha, Jennifer L. Temple, Brian M. Clemency, David Hostler

**Affiliations:** 1Department of Exercise and Nutrition Sciences, University at Buffalo, Buffalo, NY 14214, USA; 2Department of Population and Public Health Sciences, University of Southern California, Los Angeles, CA 90007, USA; 3Department of Foods and Nutrition, University of Prince Edward Island, Charlottetown, PE C1A 4P3, Canada; 4Department of Emergency Medicine, University at Buffalo, Buffalo, NY 14203, USA

**Keywords:** emergency medical services, EMS, eating patterns, nutrition, paramedics, nutrition related disease

## Abstract

Emergency medical service (EMS) providers experience demanding work conditions in addition to shift work, which increases risk for nutrition related chronic disease such as metabolic syndrome, diabetes, obesity, and cardiovascular disease. The high stress, emergent, and unpredictable nature of EMS may interfere with healthy eating patterns on and off shift, however little is known about how these conditions impact dietary patterns among EMS providers. This study aimed to understand factors impacting dietary patterns through semi-structured interviews with 40 EMS providers throughout the United States. Interviews were conducted virtually via Zoom video conference. Inductive coding was used to identify themes throughout the interviews. Salient factors mentioned in the interviews included hunger, fatigue, stress, coworker influence, ambulance posting, geographical location, agency policy, and culture. Factors were grouped into 4 domains: physiological factors, psychosocial factors, physical environment, and organizational environment, represented by an adapted version of the social ecological model of health behaviors to include factors influencing eating patterns specific to EMS, which may contribute to overall health. Various barriers to healthy eating exist within EMS, and future studies should explore interventions at each level of our proposed model to improve conditions and reduce nutrition related disease risk in this essential population.

## 1. Introduction

Emergency medical service (EMS) providers, including paramedics and emergency medical technicians (EMT) provide essential services 24 h a day. They work in shifts ranging in duration from 8.5 to 24+ hours, and up to 80% of EMS providers report working multiple jobs and/or overtime hours with variable shift rotations [1,2]. Shiftwork is associated with circadian misalignment [3,4,5,6] poor sleep quality [7] and increased risk of metabolic syndrome, type 2 diabetes, obesity and cardiovascular disease [8,9,10,11,12,13]. EMS providers also experience elevated rates of these diseases relative to the general population [14,15,16].

In addition to the stress of shift work, EMS providers’ work environment is unpredictable and inconsistent, contributing to erratic sleep and eating patterns [16]. They face high levels of stress and occupational fatigue [1] and are at greater risk of work-related injury compared to the general population and other public safety professionals [15,16,17]. EMS professionals provide an essential service and are responsible for carrying out fast paced procedures including ventilating patients, administering cardiopulmonary resuscitation, administration of potentially dangerous drugs, and assisting with childbirth [18]. Maintaining an experienced, specialized workforce will contribute to the quality and efficiency of our healthcare system. It is crucial to protect the health of this work force.

Diet is a modifiable factor that can reduce metabolic disease risk [19]. Previous research reports that shift workers tend to eat more meals and snacks at later hours of the day, and their diets are higher in snacks, sweets, alcohol, sugar sweetened beverages, and lower in dietary fiber [20]. Some studies report no difference in energy intake between day and night shift workers [21,22], however, we previously reported that among EMS providers, those working the night shift consume more overall energy (kilocalories) and grams of total fat compared to providers working the day shift [23]. Work environment may be an important consideration when assessing impact of diet on health outcomes in the shift working population. Little is known about what specific barriers to healthy eating currently exist among EMS providers.

The unpredictable nature of EMS causes significant variation in daily job demands and conditions [24]. Depending on the agency, providers may remain posted in their ambulance during a shift with minimal room for food storage and no access to refrigeration [25]. Alternatively, those who reside at the station between calls may have abundant access to meals and snacks during their down time. Call volume also varies depending on the agency and the day, posing unique challenges to healthy eating and sleep patterns [16].

The social ecological model of health explains that health is impacted by the interaction between the individual, the group/community, and the physical, social, and political environments [26,27,28]. Health behaviors in the context of EMS have been described previously by researchers in Australia, where the paramedicine system differs from the United States (US) [3]. Increasingly, agencies in the US are governed by a private system and are implementing new, more efficient response systems, incurring arduous conditions for providers. This study aimed to describe the lived experience of EMS providers in the United States to understand the impact of a rapidly changing system providers’ health and overall wellbeing. 

## 2. Methods

### 2.1. Participants

A combination of purposive and snowball sampling was used to recruit participants. EMS coordinators throughout the United States were identified from their respective State’s Department of Health website. We contacted all EMS coordinators where updated contact information was available on their website. A brief overview of the study, a study flyer, and a link to the screening survey were emailed to EMS coordinators to forward to EMS providers in their county. Participants were also asked to forward the email to anyone they thought would be interested. Interested providers were asked to complete a screening survey, which inquired about inclusion and exclusion criteria, and we contacted eligible participants to schedule a 1 h interview via Zoom (Zoom, San Jose, CA, USA, Version 5.4.6). We used data from the screening survey to describe the characteristics of the final sample (Table 1). Eligible participants were 18 years or older, worked in EMS, and provided patient care as their primary duty. Of those who completed the screening survey, 3 were ineligible, 30 did not respond to the invitation to participate, 40 participated in the interview, and no one dropped out. A portion of participants were involved in a previous study in our lab, but otherwise had no relationship with the study team prior to the interviews.

### 2.2. Data Collection

This study used a phenomenological approach to understand and describe individuals’ experience working in EMS, specifically related to eating patterns. The study was approved by the University at Buffalo’s Institutional Review Board. At the beginning of the appointment, participants were told the reason for the research and informed consent was obtained and documented electronically via RedCap [29] Using an interview guide (Appendix A), a registered dietitian conducted semi-structured interviews to inquire about participants’ lived experience working in EMS. Questions were informed by previous findings from our lab [23] and designed based on the theoretical domain framework, which was developed to examine influences on behavior of health professionals [30]. The questions were specifically related to work conditions, sleep, and eating patterns. The interview lasted 45–80 min and took place over Zoom from a private location of the participants’ choice, and the interviewer was in a private room in the Nutrition and Health Research lab at the University at Buffalo. Following the interview, the interviewer took notes to document key topics discussed in the interview. The interviews were audio and video recorded and transcripts were generated by Zoom. 

### 2.3. Analysis

All transcripts were reviewed and compared to interview recordings to verify accuracy and allow for familiarization with the data prior to coding. Transcripts were coded using Atlas.ti 9 (Atlas.ti Scientific Software Development, Berlin, Germany). Inductive thematic analysis coding was employed to identify major themes in the data following the steps suggested by Braun and Clarke [31]. Forty transcripts were divided among five independent coders. Coders were trained by the study leader, including random selection of one transcript to be coded by each, followed by reconciliation of disagreements to ensure consistency across coders. The initial round of coding involved in vivo coding and informed the initial codebook, which was then applied in a second round of coding by a separate coder. Throughout the coding process, reliability was met through weekly meetings with all coders to review each transcript, allowing for full agreement among coders (inter-coder agreement = 1). Finally, similar codes were categorized into groups and through an iterative process, major themes were identified based on those groups. Using the codes and major themes, we developed a conceptual model based on the social ecological model of health to inform future interventions in this population [32].

## 3. Results

Participants were 50% female, predominantly white, with a mean age of 33 ± 10 years and mean body mass index (BMI) in the overweight range (28.2 ± 5.7 kg/m^2^). We identified common factors discussed throughout the interviews that influenced eating patterns. The factors could all be grouped into the following overarching themes: physiological factors, psychosocial factors, physical environment, and organizational environment. These themes informed a conceptual model based on the social ecological model of health to reflect the interrelated factors influencing eating patterns among EMS providers (Figure 1). Within each theme, factors uniquely experienced by EMS providers included hunger, fatigue, stress, coworker influence, ambulance posting, geographical location, agency policy, and culture. We discuss these factors below. 

**Table 1 nutrients-14-04884-t001:** Participant Characteristics.

	N/Mean ± SD	%
**Job title**		
Paramedic	23	57.5
EMT *	9	22.5
Flight paramedic	2	5
AEMT-CC **	2	5
Firefighter	3	92.5
Volunteer	1	2.5
Age	33 ± 10.8	
Sex	20 female	50
BMI (kg/m^2^)	28.2 ± 5.7	
**Race**		
American Indian or Alaskan Native	0	0
Asian	1	2.5
White	40	100
Black or African American	1	2.5
Native Hawaiian or other Pacific Islander	0	0
More than 1 race	2	5
**Ethnicity**		
Hispanic or Latino	1	2.5
Not Hispanic or Latino	39	97.5

* EMT = Emergency Medical Technician. ** AEMT-CC = Advanced Emergency Medical Technician-Critical Care.

### 3.1. Physiological Factors

Physiological factors included fatigue and hunger. Participants reported being too tired to cook after work and were more likely to rely on convenience food during a shift and on days off. Participants mentioned that their food choice differed when they were tired compared to when they were well rested, and some mentioned eating as a tool to help them stay awake during a shift. 


*Participant #1: I definitely noticed when i’m like super tired, then I would eat like much more poorly so I definitely go for more like sugary things and coffee, and so it kind of depends on how intense of a workload I have how tired, I am.*



*Participant #2: And i’ll eat to stay awake while I am doing [patient notes], and I would love to tell you that eat carrots right, but instead I am eating ranch doritos.*


Hunger level was mentioned by participants as influencing their food choice. Participants mentioned having to go long periods of time without eating during work and that their hunger level affected their mood. 


*Participant #3: Like you’re so busy that you forget to eat and that’s like something that happens, like so often like it just it becomes like your secondary priority.*


### 3.2. Psychosocial Factors

Psychosocial factors included stress and coworker influence. Participants discussed stress related to call volume, call severity, COVID-19 precautions, sleep deprivation, and their work schedule. Stress often led to eating more and was discussed as a factor impairing ability to plan and prepare meals ahead of time. Planning was a factor that facilitated eating during work, and those who did not plan relied on acquiring food during their shift, which was unpredictable. The effort involved with planning was discussed as a barrier to consistent meal preparation. 


*Participant #4: If it’s like an overly stressful week or something…Sometimes you eat more just to like stress eat pretty much.*



*Participant #5: So right now I don’t really have the time to actually sit down and think about and plan out a meal for the week this just grab and go for right now until my life gets a little bit less stressful.*


In some, stress leads to decreased appetite and reduced energy intake resulting in unintentional weight loss.


*Participant #6: It just got to the point, I was getting so stressed, I just couldn’t eat you know it’s like now it’s like I have to force myself to eat.*


Providers typically work in crews of at least two on an ambulance, and report that their partner influences their food choice. Others report being influenced by coworkers at the base. Those who worked at fire-based agencies discuss eating together as part of the culture, and that crews often cooked dinners together.


*Participant #7: There’s definitely like that peer pressure like hey we’re putting in a pizza order, what do you want…so I feel like I eat out way more [at] work than I ever would at home, we very rarely eat out at home.*


### 3.3. Physical Environment

Factors that influenced the physical environment included geographic location, and the use of ambulance posting. Providers stated that during a shift, their food choice was dependent on availability, which varied based on location and base conditions. Providers working in urban areas had access to more food options, however, they often have a higher call volume, which they mentioned as a barrier to eating due to lack of time, inability to return to the base between calls, and limited options available in the ambulance. Some hospitals have breakrooms with snacks and beverages for providers, but availability is unreliable, and variety is limited. Food availability was particularly problematic for those who worked the night shift, so they often rely on fast food or gas stations.


*Participant #8: So your choices get incredibly limited at night there’s a there’s a nice … chain here called price chopper market 32 … where they have a nice food court and you know place where you can get stuff but that closes at 7 pm…So most of the time it’s just a gas station or 711 or some of the fast foods do late night.*



*Participant #9: And then, if we ran night calls all night, I would just stop you know mcdonald’s is like the only thing open for some reason everything else in [location] shuts down early so. That kind of be our go to … that and gas stations.*


Returning to a base between calls was discussed as a facilitator to eating during a shift compared to ambulance posting, which was discussed as a barrier. Having a base increases availability of snacks and ability to cook, store, and reheat food during a shift. However, call volume still precludes cooking during a shift in many cases. The potential to be interrupted by a call was mentioned by providers as a barrier to cooking and purchasing food and led to preference for convenient food options.


*Participant #10: But when you’re trying to cook something that was our biggest problem at [agency], we would do like spaghetti dinners and stuff, we get everything all cooked and all of a sudden you’re getting ready to eat and a call would go out.*



*Participant #11: Well we’re lucky, because we can we can go back to the station… and we have a refrigerator which is awesome, a refrigerator and a microwave, which is another factor to what’s so nice there, because if you work in an ambulance you don’t have access to a microwave or refrigerator and you know working 12 h days ambulance could get hot too...*


### 3.4. Organizational Environment

The organizational environment includes factors specific to the EMS system that influence eating patterns such as agency policy and culture. Some policies in place to increase productivity impair ability to eat during a shift. Providers often do not have an official meal break and workload was mentioned as a common barrier to eating. 


*Participant #12: There’s no like 30 min lunch break you’re just working throughout the entire day you’re always doing something; training, doing paperwork, going on calls… it’d be nice if we were allowed to, but when there’s only two people at the station, you and your partner and that’s it covering the entire town, you really can’t take a break when you want to so you can’t have a set time where you’re not working and you can dedicate that time to having a nice healthy meal. You just can’t do it.*


EMS culture was influenced by the agency organization, management, and coworker support, and discussed by providers as a factor influencing eating patterns. Unsupportive management was discussed as a factor preventing eating during a shift leading to burnout and high rates of employee turnover. 


*Participant #2: yeah and there’s a lot of pressure to take open shifts and it’s hard to not feel guilty, because you want to help your coworkers right, because if you don’t take that open shift that’s more work on them*



*Participant #13: yeah I don’t know, and I think they just work, so much so, they don’t have time to work out or like care for themselves, like mentally or physically… There’s like no gym there’s no like incentive for anybody to work out…*



*Participant #14: yeah the stress level can be elevated because of the business…we got supervisors breathing down our necks sometimes you know, trying to make us clear, so we can just go take another call..*


## 4. Discussion

EMS providers experience increased risk of chronic disease compared to the general population [33,34]. The purpose of this study was to understand factors related to eating patterns among EMS providers based on their lived experiences by conducting semi-structured interviews with providers throughout the United States. We expanded previous findings by describing each theme in the context of EMS, specifically within the United States. We also developed a conceptual model based on the social ecological model of health to inform future interventions in this population.

The social ecological model of health was developed based on the understanding that health is affected by the interaction between the individual, the community, and the physical, social, and political environment [26,35]. It acknowledges the role of context in the development and treatment of health problems. We adapted the social ecological model to include factors specific to EMS at each of the following levels: physiological, psychosocial, physical environment, and organizational environment. Ideally, interventions in this population should include action at each level to improve eating patterns and overall health, however, operations level changes may be difficult to achieve in this profession.

Physiological factors included fatigue and hunger. Previous research in animals and humans report that food intake increases after sleep loss [36,37]. Fatigue is a common problem noted among EMS providers [1], and previous qualitative work in nurses [38] report fatigue as a factor hindering ability to prepare healthy meals. Fatigue may interfere with cooking, meal planning, and lead to reliance on convenience foods in addition to increasing preference for calorie dense, high carbohydrate foods [37]. In this study, providers reported that hunger level influenced food choice and preference. Sleep loss [39], in addition to short term food deprivation when providers are too busy to eat, may lead to overcompensation of energy intake. Interventions at this level should acknowledge that fatigue may interfere with cooking and meal planning throughout the week, so thoughtful use of nutritionally balanced, easy to prepare meals and snacks should be encouraged. Providers should also receive information regarding the impact of fatigue, what to expect, and how to address it, to give them tools to overcome this unavoidable aspect of the job. 

Psychosocial factors include the individual effect and the interaction between psychological and social influence. We identified stress and peer influence as common psychosocial factors. Stress impacts food choice through neurocircuitry and hormonal pathways [40] and has been associated with intake of more pro-inflammatory diets [41]. It is also known that eating is strongly influenced by social context [42]. These factors influence eating patterns and may be an opportunity for intervention by using social interactions to support healthy dietary patterns. Interventions may include group nutrition education classes or training specific to the conditions in EMS. Group nutrition interventions have been successful at improving mental health symptoms, including stress, among participants who consumed a Mediterranean diet high in vegetables, legumes, nuts, and fruit [43]. This type of intervention may be beneficial in EMS providers.

The physical environment was shaped by agency location and ambulance posting. These challenges vary significantly between agencies and may not be modifiable depending on the type of EMS agency and the terms of the contract with the municipality. However, agencies may consider making nutrient dense snacks available at the base or in the ambulance. Future studies should investigate the impact of food deprivation on alertness and productivity, as this could support allocation of funds toward employee nourishment. Work environment directly and indirectly impacts employees’ health behaviors [44], and due to the unique, inconsistent physical environment in EMS, further studies should address the impact of environment on eating patterns. Specifically, studies should consider the impact of ambulance street posting on the health and wellbeing of providers. 

The organizational environment was shaped by work culture and agency policy. The collective attitudes toward eating and other health behaviors, lack of meal breaks, and intense work and schedule demands influenced eating patterns. Managerial support has been reported to reduce the risk of mental health problems [45], which may improve health behaviors [46,47]. Education at the management level to cultivate a positive work environment may reduce turnover. Further, previous studies in paramedics have found that those who have more control over their schedule are able to rest and recover better from demanding situations [48]. Sleep deprivation and fatigue in EMS leads to increased injury rates, compromised patient care, burnout, and poor health for providers, and increases employee turnover. Schedules designed to allow for adequate recovery between shifts could reduce fatigue, and impact food choice and health outcomes among providers and patients. The approach will differ based on agency type, which should be considered when designing an intervention.

Increased strategies for achieving adequate nutrition would improve employees’ health and quality of care as providers report severe hunger levels affecting patient care. Providers struggle with barriers to healthy eating during a shift and targeting these barriers in an intervention may improve adherence to lifestyle modifications. Providers are often in situations where they must rely on convenience foods that are low in nutrients and high in energy, and when paired with high stress and sleep deprivation, this may increase metabolic and cardiovascular disease risk [49,50,51]. Interventions should address factors at each level to improve employee wellness and retention. 

This study had several strengths including the nationally representative sample. This allowed us to examine the experience of EMS providers who respond to calls in various locations including in urban, rural, and backcountry areas. Our participants also worked various schedules ranging from 8–144 h (six 24h shifts in a row), days, nights, and a combination of both. The diversity of our sample, in terms of experience, allowed us to understand what factors were specific to EMS, and those that varied based on location. We were also able to understand the challenges associated with different schedule schemes. However, we were not able to quantify these differences, which is one limitation, and something that should be investigated in future studies with the goal of improving conditions to promote the health status of EMS providers. Another limitation of this study is the lack of ethnic and racial diversity. According to the 2020 National EMS Assessment, providers identifying as Asian, Black, African American, American Indian, and Alaska Native make up 17.7% of the workforce throughout the US [52], and future studies must focus on representation of all racial and ethnic groups. This was a major limitation of this study. 

## 5. Conclusions

The overall rate of turnover in EMS is 10.7% and the median cost of turnover is $86,452.05 for paid agencies [1]. These data are over 10 years old and has likely increased over that time as medical calls have gone up by 32% between 2010 and 2019 (NFPA Fire Experience Survey). The data from the current study, combined with earlier quantitative work, suggest that more work is needed to improve conditions for EMS providers to allow for a sustainable lifestyle and prevention of chronic disease, burnout, and employee turnover. Future studies should build on this work by testing interventions aimed at increasing healthy eating patterns, adjusting shift schedules to improve sleep quality and allow for recovery between shifts, and reconsidering organizational policies. These strategies could improve health and wellbeing among EMS providers and reduce employee turnover costs for agencies and physical health and increase the quality of life for EMS providers. 

## Figures and Tables

**Figure 1 nutrients-14-04884-f001:**
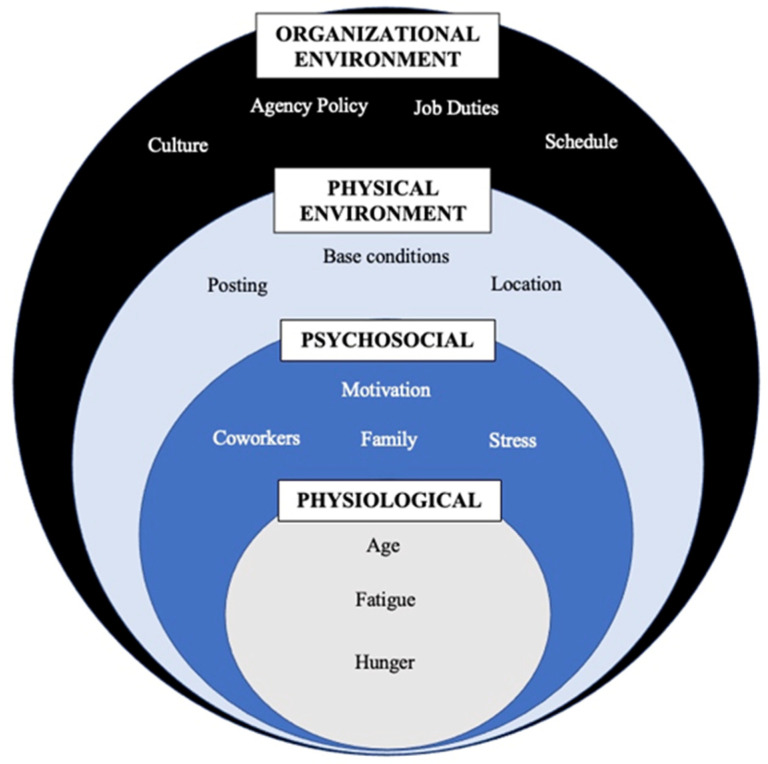
**Conceptual model**—From the major themes, we developed the following conceptual model informed by the social ecological model. The model displays the four overarching, interrelated themes, each containing commonly discussed factors describing the context in which food choice and eating occur for EMS providers.

## Data Availability

The transcripts presented in this study are available on request from the corresponding author. The transcripts are not publicly available due to privacy restrictions.

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
