# Peer review of "Eating Patterns among Emergency Medical Service Providers in the United States: A Qualitative Interview Study"

_nutrients, 2022, doi:10.3390/nu14224884_

Round 1

Reviewer 1 Report

Dear authors,

I found the topic of your article interesting and pertinent both for research and practice, and I enjoyed reading it. I have some comments and reflections that I hope that helps to improve your paper.

I hope to contribute positively to your work.

Introduction:

1.      Overall, the introduction describes the topic very well and the authors convinced me of the relevance of the present study.

2.      You have different font sizes along the introduction.

3.      Line 56-61: It lacks references.

4.      I would like to see more about the propose of the study (e.g., innovation, why it is important for future work).

Materials and methods:

1.      Overall, I think that your methods are well described.

2.      Line 83-85: The previous study that some participants participated in before is about what? Could previous participation influence participants' responses to interviews?

3.      I appreciated that you included the interview guide in the manuscript.

4.      Table 1 – I suggest that you put “job title”, “race”, and “ethnicity” in bold to be easier to the reader. Moreover, I suggest including a note with the meaning of the abbreviations.

5.      Why do you only use an inductive approach? I.e., If you have the social ecological model of health as theoretical framework and there are a similar study in Australia, why you do not used these as guide for initial codes?

6.      How do you trained the different coders to guarantee the reliability of data analysis?

7.      Do you have a second coder for each interview? Do you calculate the inter-observer agreement?

8.      Do you followed the six steps of thematic analysis suggested by Braun and Clarke?

Results:

1.      Which means the BMI of the sample? It is healthy? Overweight?

2.      Your results are very interesting. I appreciated the way in which you described.

3.      I would like to see a table with frequencies of each theme and code.

Discussion:

1.      I think the results are well contrasted with the literature.

2.      I would like to see more regarding implications for practice (e.g., suggestions for interventions).

3.      I would also like to see limitations of your study.

Apart from these comments, I consider your article could be valuable for the research community.

Author Response

Below I have included the reviewer's comments in bold, followed by our response in plain text. I want to thank the reviewer for thorough and thoughtful comments that has certainly strengthened this paper.

Introduction:

  1. Overall, the introduction describes the topic very well and the authors convinced me of the relevance of the present study. – Thank you.
  2. You have different font sizes along the introduction. – Thank you for pointing this out, it has been fixed
  3. Line 56-61: It lacks references. – Apologies for the oversight, we have added references
  4. I would like to see more about the propose of the study (e.g., innovation, why it is important for future work). – Thank you for this comment, we have added information to the introduction to outline the need for this work.

Materials and methods:

  1.  Overall, I think that your methods are well described. Thank you.
  2. Line 83-85: The previous study that some participants participated in before is about what? Could previous participation influence participants' responses to interviews? Our previous study examined dietary intake via mobile food phone records among EMS providers on 2 days on and 2 days off shift. Participation in this study would not influence participants’ response in this study, however we included this to note that some participants had met the interviewer previously while others had not. There is no reason to believe participation in the previous study would influence participants’ responses to interviews, this was only included due to recommendation from the COREQ checklist (consolidated criteria for reporting qualitative research). In the COREQ checklist, under the research team and reflexivity domain, there is a section on relationship with participants, which asks whether a relationship was established prior to study commencement.
  3. I appreciated that you included the interview guide in the manuscript. Thank you for your appreciative comments, they mean a lot!
  4. Table 1 – I suggest that you put “job title”, “race”, and “ethnicity” in bold to be easier to the reader. Moreover, I suggest including a note with the meaning of the abbreviations. Thank you for these suggestions, I have updated the table accordingly.
  5. Why do you only use an inductive approach? I.e., If you have the social ecological model of health as theoretical framework and there are a similar study in Australia, why you do not used these as guide for initial codes? Great question. Due to the phenomenological approach we chose for this paper, we wanted to focus on the participants’ lived experience, and did not want the previous findings to influence what we learned from this study.
  6. How do you trained the different coders to guarantee the reliability of data analysis? All coders were trained by the study leader. Then, all coders independently coded one randomly selected transcript, then as a group, all coders went through the transcript to reconcile any inconsistencies between individual coders. Throughout the coding process, reliability was ensured through weekly meetings with all coders to review each transcript. Each transcript was coded by 2 separate coders at different stages, and the study leader reviewed all transcripts.
  7. Do you have a second coder for each interview? Do you calculate the inter-observer agreement? There were two separate coders for each transcript. We did not calculate inter-observer agreement due to our qualitative, thematic approach to coding. All coders met weekly to discuss the codes for each transcript. After the first round of coding all 40 transcripts, we reviewed the codes to develop the initial code book. Then all transcripts were coded by a different coder for the second time using the initial codebook. This allowed all coders to become thoroughly familiar with all the transcripts, discuss all the themes, and code consistently.
  8. Do you followed the six steps of thematic analysis suggested by Braun and Clarke? Yes, we did follow these steps.
  9. Which means the BMI of the sample? It is healthy? Overweight? The mean BMI was 28.2, which is considered overweight.
  10. Your results are very interesting. I appreciated the way in which you described. Thank you! Your comment is deeply appreciated.
  11. I would like to see a table with frequencies of each theme and code. We do have a table and I am happy to provide this, however the reason it was left out of the paper was due to the qualitative nature. We felt that reporting the data in a quantitative way would be misleading since the methods were purely qualitative, and the data are not empirically generalizable. Braun and Clarke caution against utilizing “counts” for codes, as it suggests reliance on prevalence and accuracy for development of themes, which is not true for a qualitative analysis.

Discussion:

  1. I think the results are well contrasted with the literature. Thank you.
  2. I would like to see more regarding implications for practice (e.g., suggestions for interventions). Thank you for this comment, we have added suggestions for interventions.
  3. I would also like to see limitations of your study. Thank you for catching this, we have added limitations to the discussion.

Reviewer 2 Report

This paper purposed to describe the lived experience of EMS providers in the United States to understand the impact of a rapidly changing system provider’s health and overall wellbeing. I do have some comments as listed below in the order noted.

Comment 1: Please provide the statements with an overview of eating patterns among emergency medical service providers for better understanding of the study.

Comment 2: The quality of the participants is very important, especially in a qualitative interview study. For this reason, please clarify Sampling Strategy and Recruitment in the Methods section.  

Comment 3: Please also provide the Development of Interview Guides & Semistructured Interviews in the Methods section.

Comment 4: Please provide the intercoder reliability tests in the Analysis subsection.

Author Response

Below I have included the reviewer's comment in bold followed by our response in plain text. I want to thank the reviewer for their thorough comments and for taking the time to review this paper. We are very grateful for the opportunity to share our work, and these comments have strengthened this paper.

Comment 1: Please provide the statements with an overview of eating patterns among emergency medical service providers for better understanding of the study. I truly appreciate this comment, but I don’t quite understand. Are you referring to the participant statements? If so, do you mean add more about EMS providers eating patterns in the results section? I apologize for misunderstanding, but if you could clarify this comment I would greatly appreciate it!

Comment 2: The quality of the participants is very important, especially in a qualitative interview study. For this reason, please clarify Sampling Strategy and Recruitment in the Methods section.  Thank you for this comment! The methods did need clarifying, we updated the recruitment strategy and hope it is clearer. Please let us know if more information would be helpful.

Comment 3: Please also provide the Development of Interview Guides & Semistructured Interviews in the Methods section. We have added information on the development of the interview guide and semi-structured interviews in the Methods section under “Data Collection”

Comment 4: Please provide the intercoder reliability tests in the Analysis subsection.We did not calculate inter-observer agreement due to our qualitative, thematic approach to coding. All coders met weekly to discuss the codes for each transcript. After the first round of coding all 40 transcripts, we reviewed the codes to develop the initial code book. Then all transcripts were coded by a different coder for the second time using the initial codebook. This allowed all coders to become thoroughly familiar with all the transcripts, discuss all the themes, and code consistently.

Round 2

Reviewer 1 Report

Dear authors,

I am very happy with your answers and the improvement of your paper, and glad that you considered my suggestions. I only have some comments and reflections left.

Comment: How do you trained the different coders to guarantee the reliability of data analysis? All coders were trained by the study leader. Then, all coders independently coded one randomly selected transcript, then as a group, all coders went through the transcript to reconcile any inconsistencies between individual coders. Throughout the coding process, reliability was ensured through weekly meetings with all coders to review each transcript. Each transcript was coded by 2 separate coders at different stages, and the study leader reviewed all transcripts.

New comment: Thank you for the clarification. I suggest that you clarify this information on your paper.

Comment: Do you have a second coder for each interview? Do you calculate the inter-observer agreement? There were two separate coders for each transcript. We did not calculate inter-observer agreement due to our qualitative, thematic approach to coding. All coders met weekly to discuss the codes for each transcript. After the first round of coding all 40 transcripts, we reviewed the codes to develop the initial code book. Then all transcripts were coded by a different coder for the second time using the initial codebook. This allowed all coders to become thoroughly familiar with all the transcripts, discuss all the themes, and code consistently.

New comment: I understand your explanation. However, you can calculate the inter-observer agreement in thematic approach. It will help to prove the reliability of your data. See: https://help-nv11.qsrinternational.com/desktop/procedures/run_a_coding_comparison_query.htm

Comment: Do you followed the six steps of thematic analysis suggested by Braun and Clarke? Yes, we did follow these steps.

New comment: Please, add this information in your method.

Comment: Which means the BMI of the sample? It is healthy? Overweight? The mean BMI was 28.2, which is considered overweight.

New comment: I think you should mention this in your article. It gives more relevance to the study – If most professionals have overweightness, then it is urgent to intervene.

I suggest the publication of this article after addressing these comments. Congratulations on your paper! Thank you, obrigada.
